

# Analysis of adult damselfly fecal material aids in the estimation of antibiotic-resistant *Enterobacterales* contamination of the local environment

Yuyu Yamaguchi[1,*], Torahiko Okubo[2,*], Mizue Matsushita[2], Masashi Wataji[1], Sumio Iwasaki[3], Kasumi Hayasaka[3], Kouzi Akizawa[3], Junji Matsuo[2], Chikara Shimizu[3] and Hiroyuki Yamaguchi[2]

[1] Hokkaido Sapporo Asahigaoka Senior High School, Sapporo, Japan
[2] Department of Medical Laboratory Science, Faculty of Health Sciences, Hokkaido University, Sapporo, Japan
[3] Hokkaido University Hospital, Sapporo, Japan
[*] These authors contributed equally to this work.

## ABSTRACT

Because damselflies are ubiquitously but focally present in natural environments and play a critical role as predators of other insect species, the fecal matter of damselflies may be useful for investigating antibiotic-resistant bacterial populations, including human pathogens, in local environments. We therefore examined the prevalence of antibiotic-resistant bacteria, including *Enterobacterales*, in fecal material from 383 damselflies (adults and larvae) collected from seven locations around Sapporo City, Japan, in 2016 and 2017. Fecal samples were plated on soybean casein digest (SCD) agar plates with and without antibiotics (SCD-A and SCD-w/o, respectively) to identify environmental bacteria and gut bacteria, respectively, and on MacConkey agar plates with antibiotics (MacConkey-A) to select for Gram-negative bacteria, including human pathogenic *Enterobacterales* species. The prevalence of colonies on each of the plates was compared, and representative colonies on MacConkey-A plates were identified to the species level using an API 20E kit and the MALDI Biotyper system. Overall, SCD-w/o plates showed a gut bacterial load of approximately $10^8$ colony-forming units per adult damselfly or larva. There was a significant difference between the prevalence of colonies on the SCD-A and MacConkey-A plates, and a significantly increased prevalence of antibiotic-resistant bacteria on MacConkey-A plates was observed in samples collected from Shinoroshinkawa. Cluster analysis based on minimum inhibitory concentration values of 59 representative isolates from MacConkey-A agar plates revealed that samples from Shinoroshinkawa contained a higher prevalence of *Enterobacterales* than those from other sampling locations. Thus, fecal materials discharged by adult damselflies could be used in future studies as a simple tool for estimating antibiotic-resistant bacteria, including *Enterobacterales* species, in the local environment.

Corresponding author
Hiroyuki Yamaguchi,
hiroyuki@med.hokudai.ac.jp

## INTRODUCTION

The pandemic of multidrug-resistant (MDR) pathogens and their continuing spread is a growing global concern that represents an immeasurable threat to hospitals and other healthcare-associated facilities (*Magiorakos et al., 2012*; *Hawkey, 2015*; *Medina & Pieper, 2016*). The United States Centers for Disease Control and Prevention reported that there were more than two million cases of infection caused by MDR bacteria in the United States, resulting in approximately 23,000 deaths (*CDC, 2012*; *Johnson et al., 2014*). In addition, the European Centre for Disease Prevention and Control, the European Food Safety Authority, and the European Medicines Agency have identified an increase in the mortality rates associated with infections caused by MDR bacteria in both Asia and Africa (*De Beer et al., 2014*). As a result, the World Health Organization called on all member countries to promote rational antibiotic use to prevent the spread of antimicrobial resistance (*World Health Organization, 2007*). However, the lack of a comprehensive management protocol encompassing livestock, circulating food, public hygiene practices, and monitoring of natural environments has resulted in the failure to control the emergence of MDR bacteria (*World Health Organization, 2007*; *CDC, 2012*; *Magiorakos et al., 2012*; *Calistri et al., 2013*; *De Beer et al., 2014*; *Johnson et al., 2014*; *Hawkey, 2015*; *Medina & Pieper, 2016*). Hence, a "One Health" approach has been proposed, which would play a crucial role in controlling the emergence of MDR bacteria. Also, it is therefore that appropriate monitoring the emergence of MDR bacteria in particular into natural environments would be critically important for understanding spread of the bacteria, responsible for these control in public health.

Damselflies ubiquitously but focally inhabit natural environments such as rivers and ponds in most places around the world (*Bourret, McPeek & Turgeon, 2012*; *Dolný et al., 2012*; *Dolný, Harabiš & Mižičová, 2014*; *Ball-Damerow, Oboyski & Resh, 2015*). Because damselflies can be very sensitive to subtle changes in their environment, they may be a good indicator of environmental health and biodiversity (*Oliver et al., 2010*; *Kutcher & Bried, 2014*). Damselflies play a critical role as predators of other small insect species sometimes hatched out around human living sites, many of where contain a lot of microbes, including human pathogens (*Siepielski et al., 2011*). It is therefore likely that fecal materials discharged by damselflies could be used to examine populations of antibiotic-resistant bacteria, including *Enterobacterales* species, derived from humans in the local environment. To assess this possibility from the "One Health" viewpoint, we determined the prevalence of antibiotic-resistant *Enterobacterales* species in the fecal matter of damselflies captured from public places surrounding local rivers and ponds within distinct natural environments in Sapporo City, Japan.

## MATERIALS AND METHODS

### Antibiotics

All antimicrobial agents, including cefotaxime (CTX), ampicillin (AMP), kanamycin (KAN), tetracycline (TET), chloramphenicol (CHL), ciprofloxacin (CIP), and sulfapyridine (SPY), were purchased from Sigma-Aldrich. In addition, we selected representative

antibiotics that are used as the major veterinary and/or human pharmaceuticals in Japan (*Yamasaki et al., 2018*).

## Sample collection

A total of 315 adult damselflies were captured from June–August 2016 and in June 2017 using an insect net. The damselflies were collected from seven locations: Higashitonden (north latitude: 43.147353; east longitude: 141.337327), Shinoroshinkawa (north latitude: 43.120752; east longitude: 141.415881), Ohno (north latitude: 43.074279; east longitude: 141.341688), Shinoro (north latitude: 43.159162; east longitude: 141.364069), Tonneusu (north latitude: 43.169276; east longitude: 141.408628), Gotenzan (north latitude: 43.048401; east longitude: 141.258540), and Yasuharu (north latitude: 43.144301; east longitude: 141.315791). Each sampling site was located near a local river or pond with distinct natural environments in Sapporo City, Japan (Fig. 1). In specific, the most urban location was "Pond: Ohno", located within the grounds of Hokkaido University, near Sapporo Station, in the center of the city (Fig. 1. See "Green density"). In contrast, the "River: Gotenzan" sampling site was located in a forest and was therefore the most "natural" of the sampling locations (Fig. 1. See "Green density"). Sixty-eight larvae were also collected from three of the locations (Shinoroshinkawa, Tonneusu, and Yasuharu) in June 2017 (Fig. 1). Environmental temperatures at each of the sampling locations were obtained from the Japan Weather Association (http://www.tenki.jp/).

## Species determination for the captured adult damselflies

The species of each of the captured adult damselflies was determined morphologically based on distinguishing colors and patterns (lines and spots) on the body and wing (See Fig. 2A). Because of the difficulties associated with morphological species determination of larvae, the species of damselfly larvae samples could not be identified.

## Collection of fecal materials

Fecal samples were collected as per the protocol outlined in Fig. 3. Briefly, adult damselflies wrapped in clean paper, along with larvae that had been immersed in sterile phosphate-buffered saline (sPBS) after washing with sPBS to prevent contamination from the exoskeleton, were stored at room temperature for at least a week. Dead individuals were omitted from the subsequent experiments. Fecal materials discharged onto the paper or into the sPBS were collected into fresh sPBS solution and homogenized. Aliquots of the homogenized solutions were inoculated onto soybean casein digest (SCD) agar plates (BD Biosciences) minus antibiotics (SCD-w/o) as an indicator of total damselfly gut bacteria, SCD agar plates supplemented with CTX (50 mg/liter) or TET (50 mg/liter) (SCD-A) as an indicator of environmental bacteria that frequently resist to these antibiotics (*Huang et al., 2015*), and MacConkey agar plates supplemented with AMP (10 mg/liter), TET (10 mg/liter), or KAN (10 mg/liter) (MacConkey-A) as an indicator of Gram-negative bacteria, including human-derived pathogenic *Enterobacterales* species. All plates were incubated at 25 °C before colony enumeration to prevent over growth on agar plates covered by environmental bacteria characteristic of rapid growth above 30 °C temperature. Results were then expressed as colony-forming units (CFU) per individual damselfly or larva.

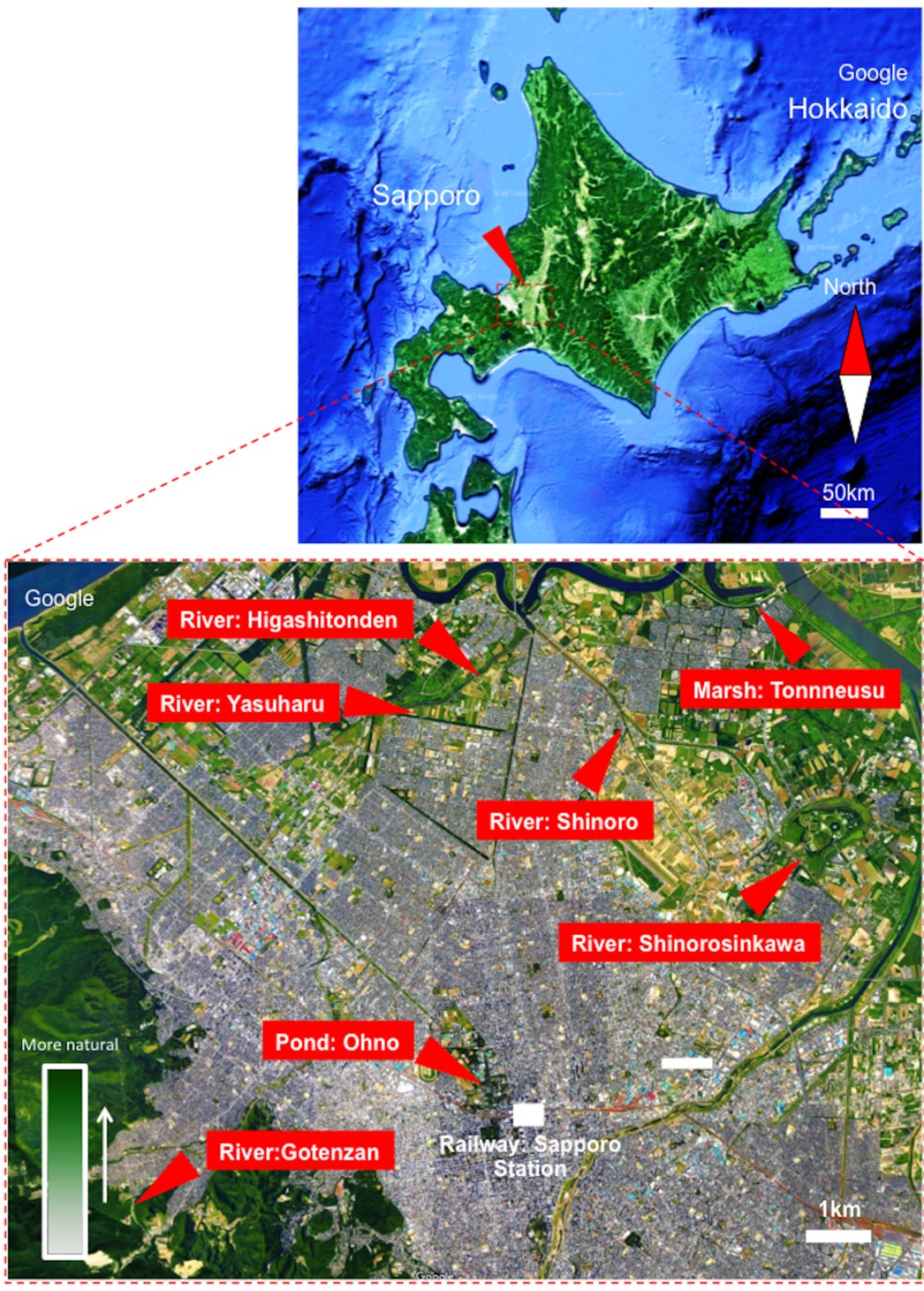

**Figure 1** **Map showing sampling locations around Sapporo City, Hokkaido, Japan.** The image was obtained from Google Maps (https://www.google.co.jp/maps). Map of Japan showing Sapporo: Image ©2018 Landsat/Copernicus, Data SIO NOAA, Navy, NGA, GEBCO, Detailed view of Sapporo: Image ©2018 Google, Data SIQ, NOAA, U.S. Navy, NGA, GEBCO, Map data.

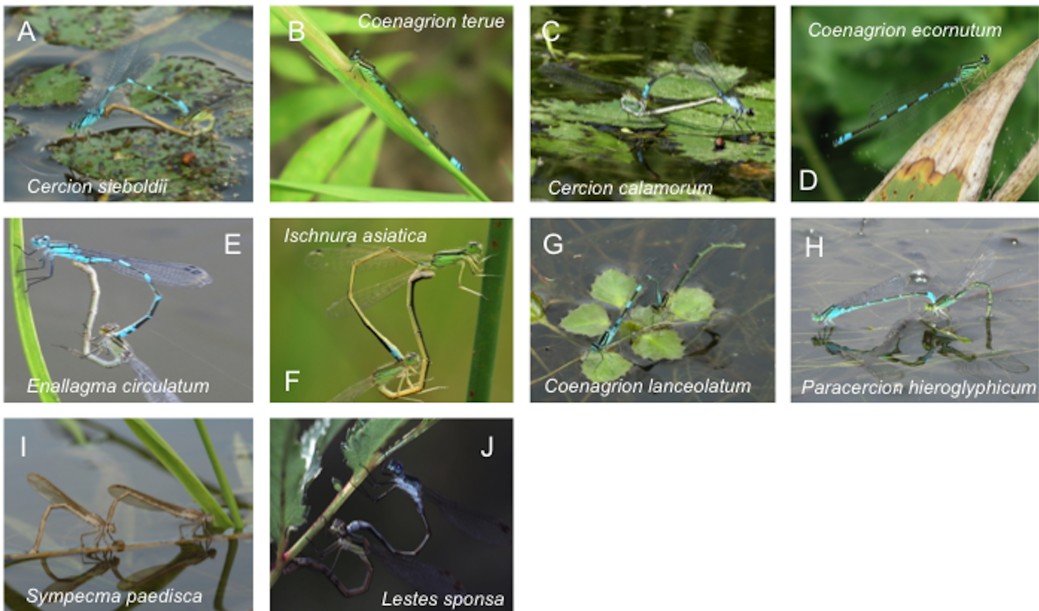

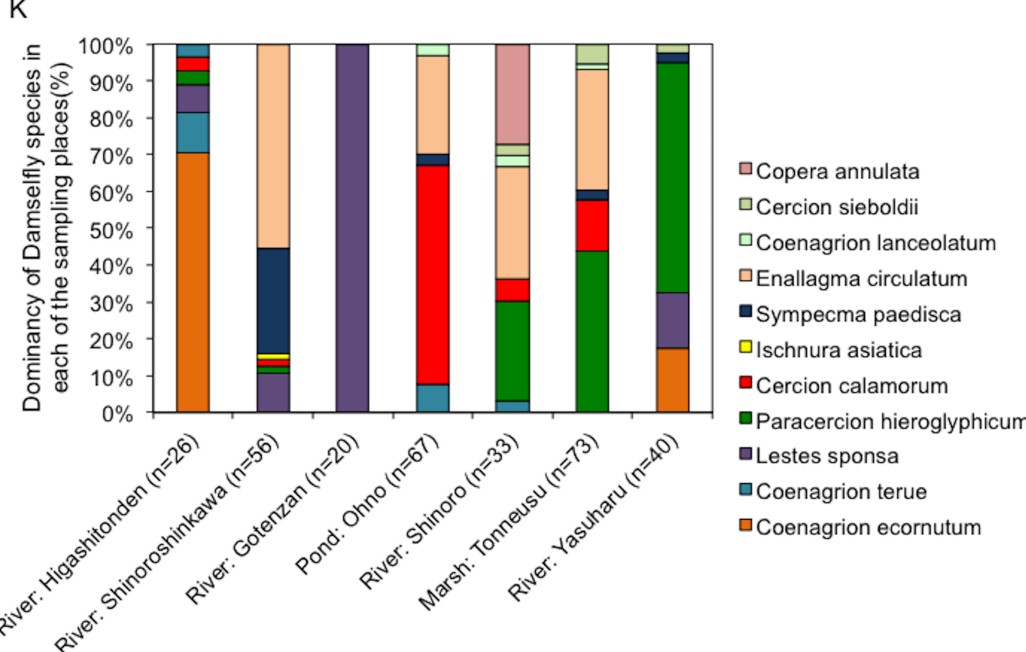

**Figure 2** **Representative images showing distinct morphological traits of damselflies and changes in the dominancy of different damselfly species at each sampling site.** (A–J) Representative images showing distinct morphological traits of damselflies. (K) The dominancy of different damselfly species differed between sampling sites, with a total of 11 species identified across all sites. The $y$-axis shows the dominancy of damselfly species as a ratio (%).

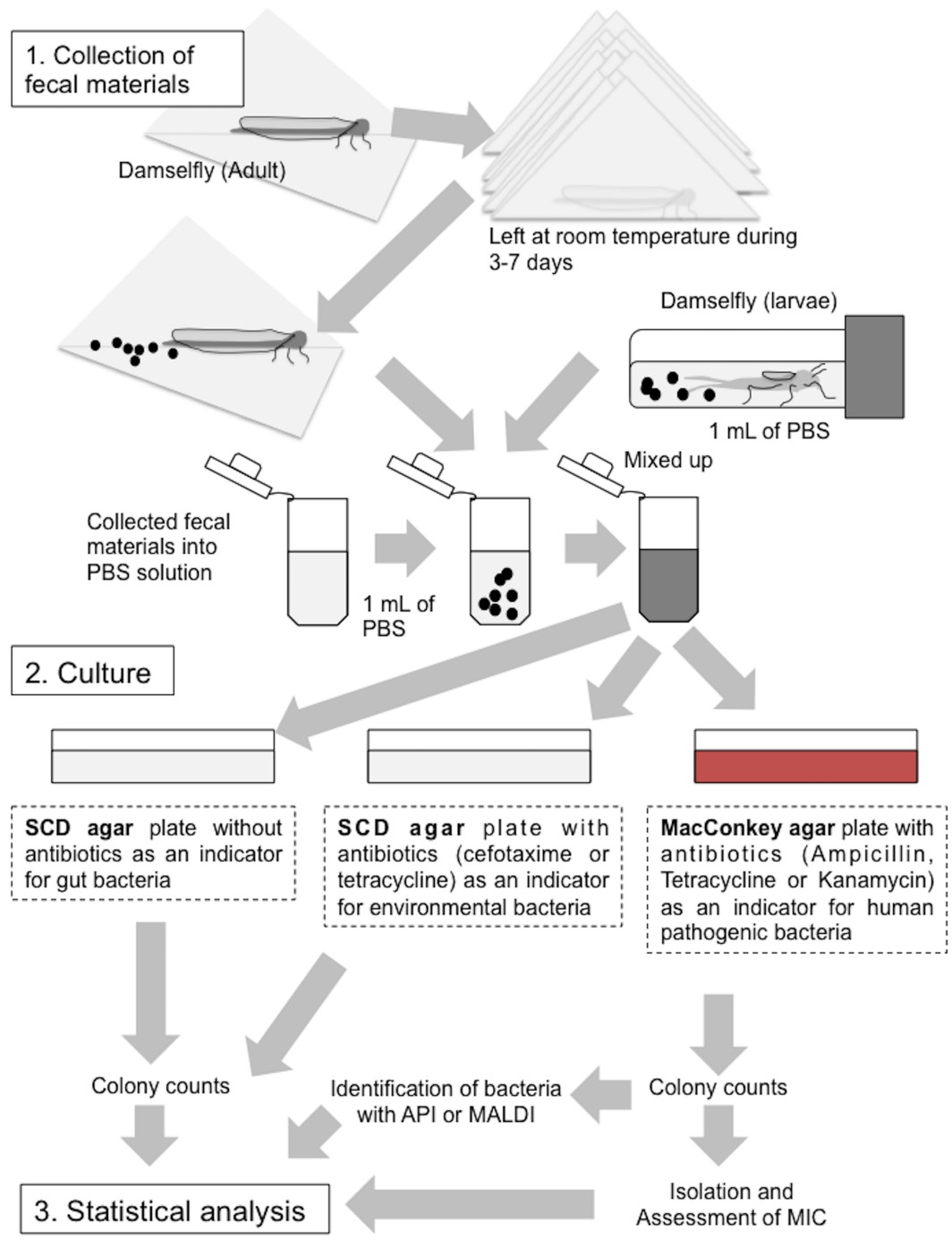

**Figure 3 Collection of fecal materials from adult and larval damselflies and cultures.** (1) Collection of fecal materials expelled by adult damselflies wrapped in clean paper, and from larvae in sPBS solution. (2) Culturing of bacteria from fecal materials on SCD agar with or without antibiotics and on MacConkey agar plates with antibiotics. (3) Statistical analysis (See ''Materials and Methods'').

Any SCD-A or MacConkey-A plate with more than one colony was considered positive for antibiotic-resistant bacteria, followed by comparing these prevalence among sampling sites; the prevalence is defined to be the number of sample showing agar plates (SCD-A or MacConkey-A plate) with more than one colony. Morphologically representative colonies on MacConkey-A plates were also isolated and used for antimicrobial susceptibility testing according to the method described below.

### Bacterial species determination

Bacterial species were identified using an API 20E kit according to the manufacturer's protocol (bioMérieux, Marcy-l'Étoile, France). The MALDI Biotyper system (Bruker Daltonics, Billerica, MA, USA) was used as per the manufacturer's instructions to identify those species that could not be determined using the API kit. Meanwhile, if the API value was low (below 80%), bacterial 16S rDNA typing was performed to define bacterial species according to the protocol described previously (*Horn et al., 1999*).

### Antimicrobial susceptibility testing

Antimicrobial susceptibility testing was performed for each of the identified bacterial isolates using the agar-dilution method to determine the minimum inhibitory concentration (MIC) values for six antimicrobial agents (AMP, KAN, TET, CHL, CIP, and SPY) on Mueller-Hinton II agar (BD Biosciences, Franklin Lakes, NJ, USA). Assays were carried out according to the criteria determined by the CLSI (*Wiegand, Hilpert & Hancock, 2008*). *Staphylococcus aureus* ATCC29213 and *Escherichia coli* ATCC25922 were used as quality control strains.

### Clustering and phylogenic analysis

Cluster analysis was performed using Cluster 3.0 for Mac OS X (Clustering Library 1.52) as described previously (*Okubo et al., 2017*). Phylogenetic trees were generated from the aligned population structures by Cluster 3.0 (see below) and visualized in Java TreeViewX (version 0.5.0). Specifically, the obtained data were transformed to a log scale, and then processed with the setting (>80% filtering and Pearson correlation (centered)). While the data associated with specific variables (temperature), total bacterial numbers on SCD-w/o, the prevalence of bacteria on SCD-A, the prevalence of bacteria on MacConkey-A were converted to an equivalent range (0–1), the MIC data were used without any alterations. The resulting data were then processed using the setting (>80% filtering and Spearman rank correlation) and converted to "cdt" files using Cluster 3.0 then visualized using TreeViewX.

### Ethics

The study reported in this manuscript did not involve any human participants, human data, human tissue, data pertaining to specific individuals, or animal experiments.

### Statistical analysis

Comparison of total CFU numbers between sampling locations was conducted using a Bonferroni/Dunn test. A *p*-value of < 0.01 was considered significant. Correlations among factors (temperature, total bacterial numbers on SCD-w/o, the prevalence of bacteria

on SCD-A, the prevalence of bacteria on MacConkey-A) were identified by Pearson's correlation coefficient test. A correlation coefficient value of $>0.5$ or $<-0.05$, with a $P$-value of $< 0.05$, was considered significant. Comparisons between the prevalence of bacteria on SCD-A and MacConkey-A agar plates were conducted using a Chi-square for independence test. Comparisons between the prevalence of antibiotic-resistant isolates obtained from MacConkey-A plates at each of the sampling locations were also conducted using a Chi-square for independence test. A combination with a $P$-value of $< 0.05$ was considered statistically significant. All calculations were conducted using Excel for Mac (2001) with Statcel3C.

## RESULTS

### Damselflies are ubiquitously but focally present in local environments

Because our study relied on the domestic behaviors of damselflies at each location, we first assessed whether focal inhabitation by adult damselflies could be observed at each of the sampling sites. A total of 315 adult damselflies were captured between June and August 2016, and in June 2017, at seven locations (Fig. 1). Morphological species identification results for the captured adult damselflies are summarized in Fig. 2A. Although all sampling sites were located within a 10-km$^2$ area, the prevalence of each of the 11 identified damselfly species varied between the sites (Fig. 2B). *Coenagrion ecornutum* was the most dominant species at the "River: Higashitonden" location, while *Enallagma circulatum* was most commonly identified at the "River: Shinoroshinkawa", "River: Shinoro", and "Marsh: Tonneusu" sites. The dominancy of *Paracercion hieroglyphicum* was higher at the "River: Yasuharu" and "Marsh: Tonneusu" sites compared with the other locations, while *Lestes sponsa* was more frequently identified at the "River: Gotenzan" site compared with all other sampling sites. In addition, 68 damselfly larvae were collected from three of the sampling sites (Shinoroshinkawa, Tonneusu, and Yasuharu) in June 2017. However, morphological species identification was not possible for the larvae. Thus, as expected, the population density of adult damselflies changed depending on the sampling location, indicating that damselflies ubiquitously but focally inhabit each of the local environments around Sapporo City.

### Relationship between temperature and total number of bacteria in the damselfly gut and the prevalence of antibiotic-resistant bacteria

Fecal materials from both adult and larval damselflies were collected, and the bacterial numbers were estimated according to the protocol outlined in Fig. 3. When estimated using SCD-w/o medium, the total number of bacteria per adult/larval damselfly was fairly consistent across all sampling locations ($\sim10^8$ CFU per adult/larva), except for the "Pond: Ohno" site (Fig. 4). We also assessed the relationship between the total bacterial load (SCD-w/o) or the prevalence of antibiotic-resistant bacteria (SCD-A and MacConkey-A plates) and environmental temperature using a Pearson's correlation coefficient test and cluster analysis with a Spearman's correlation coefficient by rank test. The results showed that temperature changes were significantly but negatively correlated with the total bacterial load on SCD-w/o plates ($r$ coefficient $-0.677$ vs. "Lowest temperature", $P < 0.0001$;

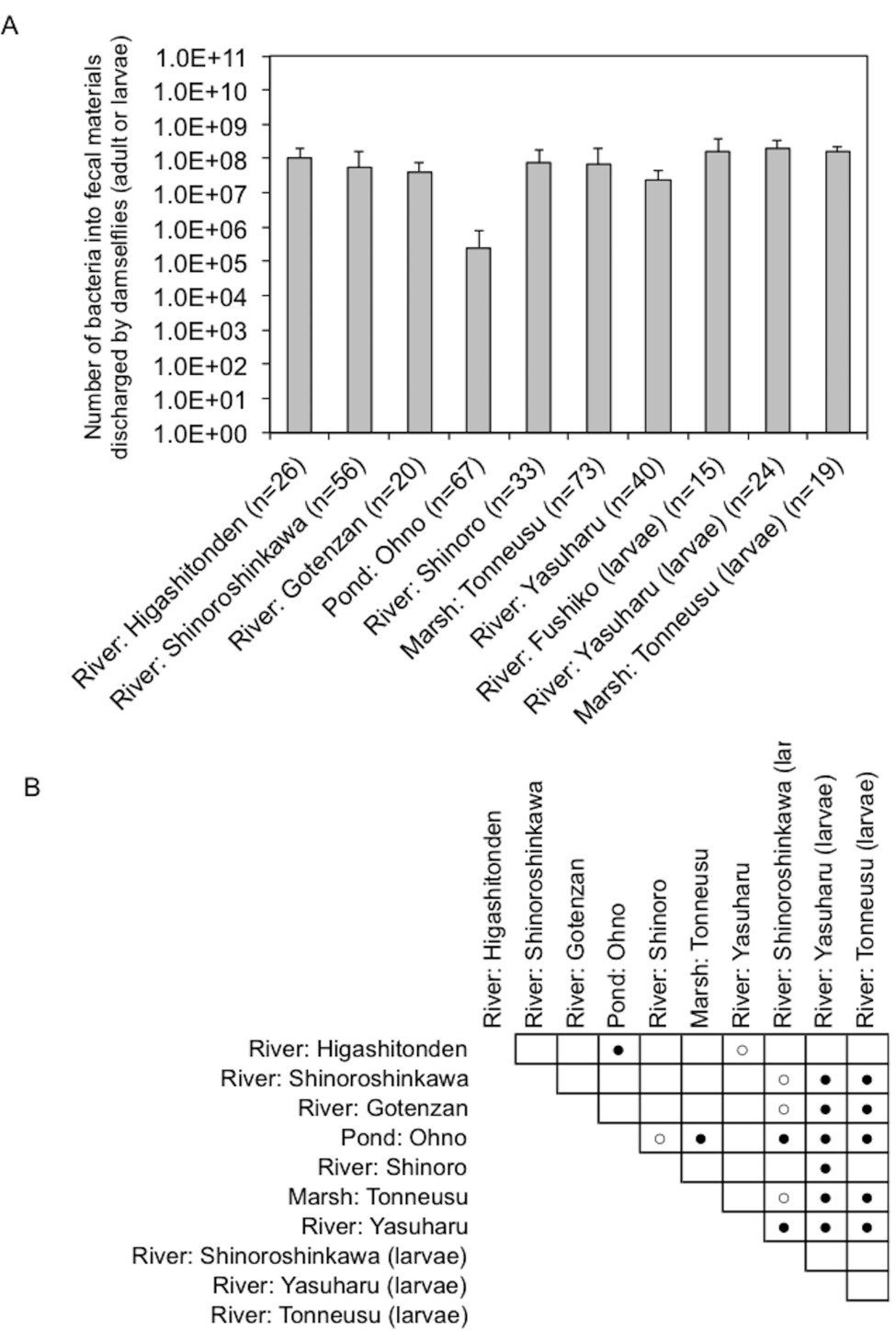

**Figure 4  Bacterial load in fecal materials estimated following culture on SCD agar plates without antibiotics.** (A) shows total bacterial numbers (CFU) in samples from adult and larval damselflies. (B) shows results of statistical analyses performed to compare results among sampling locations. Comparisons of the total CFU counts among sampling locations were conducted using a Bonferroni/Dunn test. Closed and open circles show statistical significance at $P < 0.01$ and $P < 0.05$, respectively.

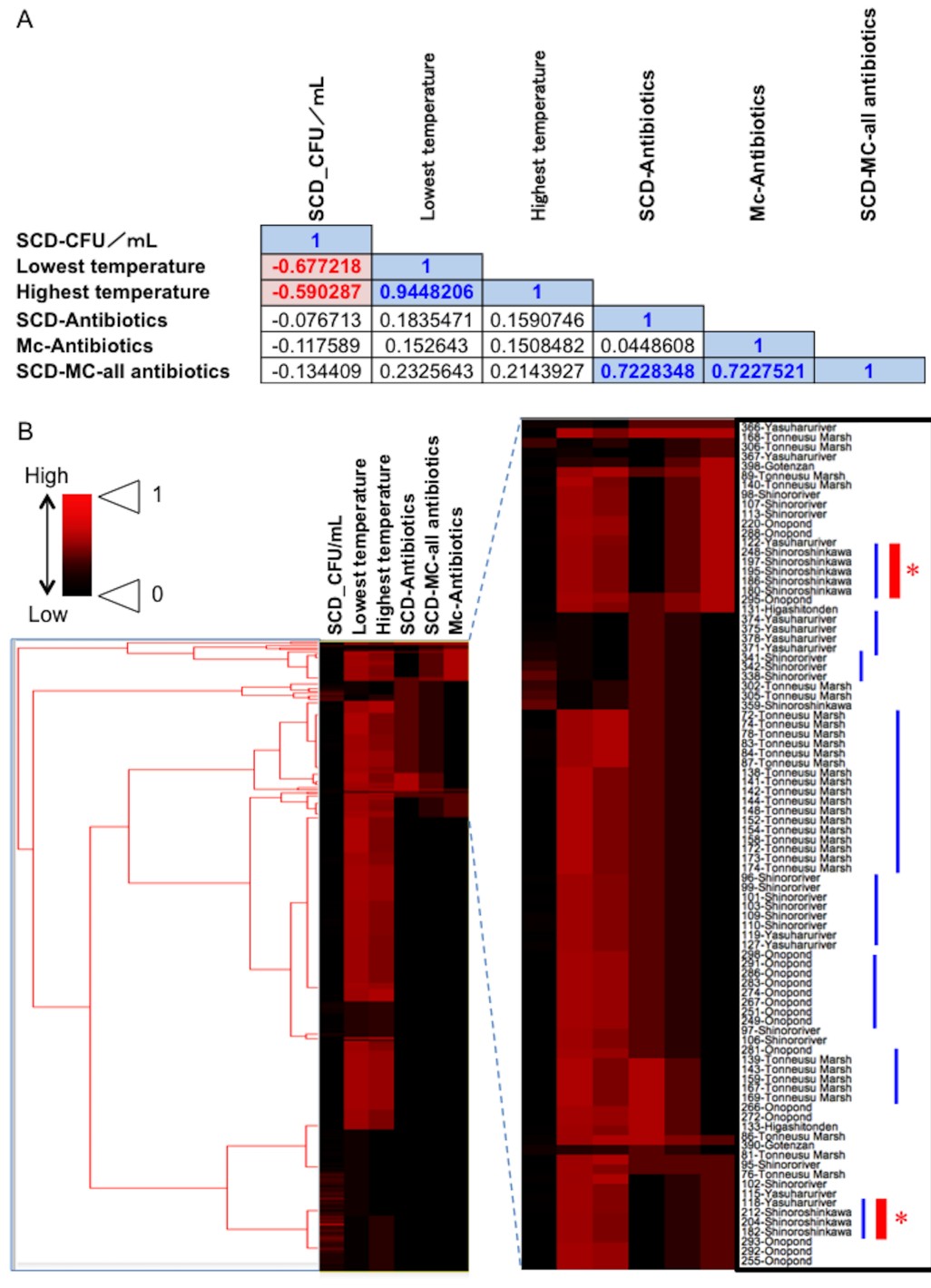

**Figure 5  Association between total bacterial load (SCD agar plate without any antibiotics) or the prevalence of antibiotic-resistant bacteria (SCD agar plates with antibiotics and MacConkey agar plates with antibiotics) and environmental temperature.** (A) Correlations among factors identified using Pearson's correlation coefficient test. A correlation coefficient value of $>0.5$ or $<-0.05$, with a $P$-value of $<0.05$, was considered significant. Blue and red indicate a positive or negative 

**Figure 5 (…continued)**
correlation with statistical significance, respectively. (B) Cluster analysis based on the above indicators (environmental temperature, total bacterial numbers in fecal samples, the prevalence of antibiotic-resistant bacteria), with several clades grouped by each of the sampling locations. Blue bars show clusters containing samples from the same location. Red asterisks show "River: Shinoroshinkawa", which had the highest frequency of emerging antibiotic-resistant bacteria. "SCD_CFU/ml" shows the number of total colonies per sample. "SCD-Antibiotics" and "Mc-Antibiotics" show the bacterial growth on SCD-A and MacConkey-A plates, respectively. Also, "SCD-Mc-all antibiotics" shows total frequency of "SCD-Antibiotics" plus "Mc-Antibiotics".

$r$ coefficient $-0.590$ vs. "Highest temperature", $P < 0.0001$) (Fig. 5A). Meanwhile, no significant correlation between temperature changes and the prevalence of colonies on SCD-A and MacConkey-A agar plates was observed. Furthermore, cluster analysis based on three different indicators (environmental temperature, total bacterial numbers in fecal samples, the prevalence of antibiotic resistant bacteria) revealed several clades grouped by sampling location (Fig. 5B, blue bars). Interestingly, samples obtained from the "River: Shinoroshinkawa" site were assigned to two different groups, which, together, had the highest prevalence of antibiotic resistant bacteria (Fig. 5B, red bars with asterisks). Thus, while the fecal bacterial load on SCD-w/o medium was fairly constant and was negatively correlated with environmental temperature, the prevalence of antibiotic-resistant bacteria changed depending on the sampling location and was not related to environmental temperature.

## Prevalence of antibiotic-resistant bacteria and identification of representative antibiotic-resistant *Enterobacterales*

To determine which antibiotic-resistant bacterial strains were most prevalent in the sampled environments, we estimated the prevalence of antibiotic-resistant bacteria in the damselfly fecal samples. SCD-A medium was used as an indicator of environmental bacteria, while MacConkey-A medium was used to select for pathogenic bacteria, including *Enterobacterales*, derived from humans. Results showed that the prevalence of environmental and human-derived pathogenic bacteria changed depending on the sampling site, irrespective of whether the samples came from adult or larval damselflies (Table S1). However, at almost all sampling locations, environmental antibiotic-resistant bacteria were significantly more prevalent than human-derived pathogenic bacteria (adult damselfly: "River: Shinoro", $P = 0.0487$; "Marsh: Tonneusu", $P < 0.0001$; larvae: "River: Yasuharu", $P = 0.0329$; "Marsh: Tonneusu", $P < 0.0001$). Interestingly, the samples obtained from the "River: Shinoroshinkawa" site uniquely showed a bias towards antibiotic-resistant human-derived pathogenic bacteria ($P = 0.0149$). Thus, the results revealed that the prevalence of antibiotic-resistant bacteria in the fecal materials of both adult and larval damselflies significantly differed between sampling sites, reflecting differences in bacterial contamination of each of the domestic environments.

Amongst the sampling locations, the "River: Shinoroshinkawa" site appeared to be statistically unique, with an increased prevalence of antibiotic-resistant *Enterobacterales* on MacConkey-A plates. To confirm this, 59 representative antibiotic-resistant
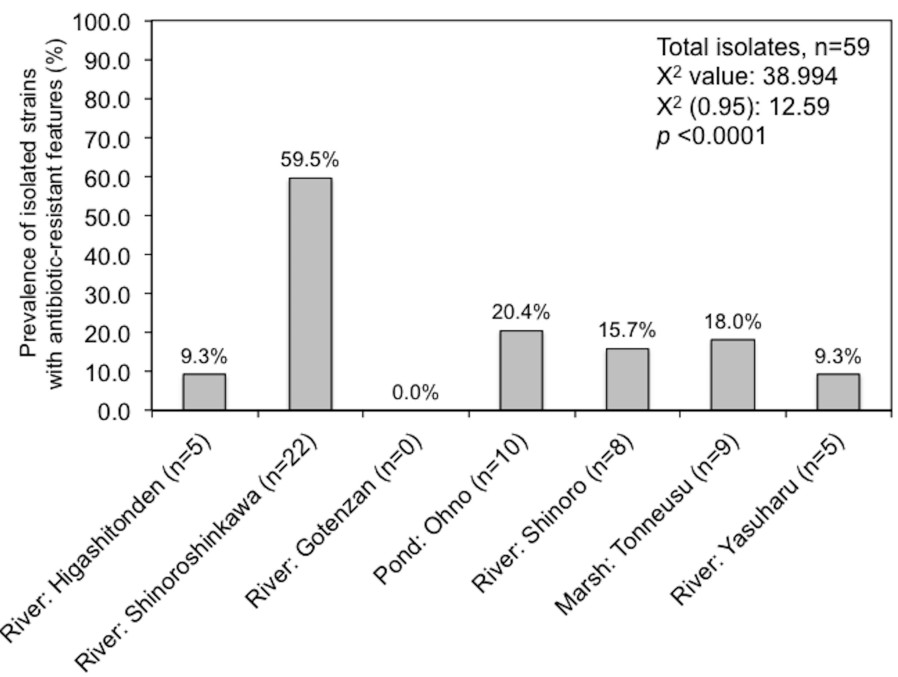

**Figure 6 Prevalence of isolated strains from MacConkey agar plates with antibiotics at each of the sample location (%).** Fifty-nine isolates were used for this study. Comparisons between the prevalence of antibiotic-resistant isolates at each of the sampling sites were conducted using a Chi-square for independence test. Antibiotic-resistant bacteria were most prevalent at the "River: Shinoroshinkawa" site, with statistical significance at $P < 0.01$.

*Enterobacterales* isolates from adult damselfly fecal material were selected from MacConkey-A plates and identified using an API kit and the MALDI Biotyper system with MIC assessments (Table S2). The predominant bacterial species identified were *Serratia fonticola*, *Serratia liquefaciens*, *Enterobacter aerogenes*, *Enterobacter cloacae*, *Klebsiella oxytoca*, and *Stenotrophomonas maltophilia*. As expected, the site with the highest prevalence of antibiotic-resistant bacteria was "River: Shinoroshinkawa" ($\chi^2$ test: $p < 0.001$) (Fig. 6). The MICs (MIC$_{50}$ and MIC$_{90}$) of AMP, KAN, TET, CHL, CIP, and SPY for each of the identified bacteria were also determined and used as a basis for clustering analysis (Table S1). Although there were no obvious differences in the MIC ranges between the sampling sites except for Shinoroshinkawa with high values of MIC90 for KAN and CIP (Fig. 7A), cluster analysis revealed that the identified bacteria derived from the "River: Shinoroshinkawa" site formed a cluster and showed higher MIC values and a greater prevalence of multidrug resistance compared with other sampling locations (Fig. 7B). Thus, the "River: Shinoroshinkawa" site was unique amongst the sampling locations, with the highest frequency of antibiotic-resistant *Enterobacterales*.

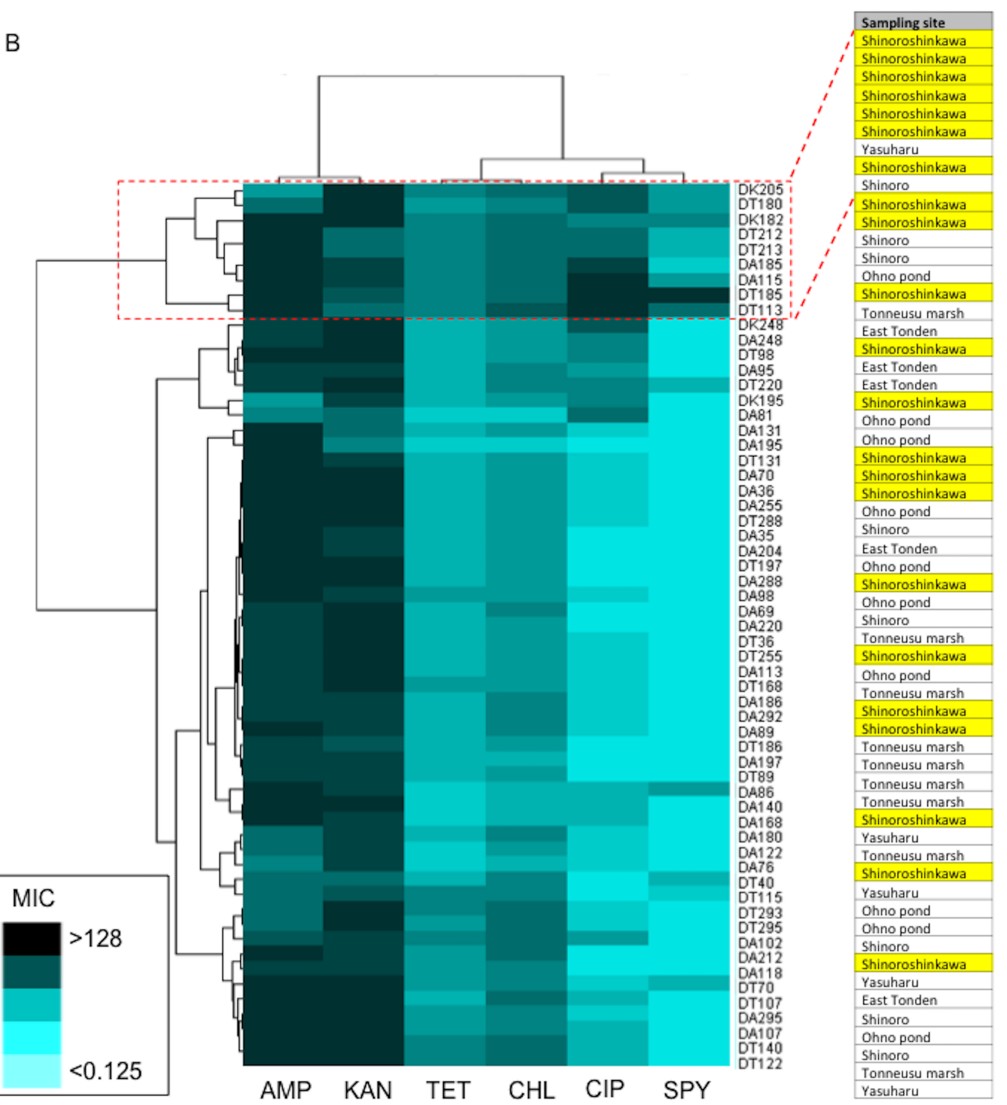

**Figure 7 Comparison of the prevalence of antibiotic-resistant bacteria on SCD agar plates with that on MacConkey agar plates, and identification of representative antibiotic-resistant *Enterobacterales*, including MIC data.** (A) Antimicrobial susceptibility testing results for each of the isolates ($MIC_{50}$ and $MIC_{90}$) for six antimicrobial agents (AMP, KAN, TET, CHL, CIP, and SPY). Isolates were tested using the agar-dilution method on Muller-Hinton II agar medium (BD Biosciences) according to the criteria determined by the CLSI (*Wiegand, Hilpert & Hancock, 2008*). (B) Cluster analysis based on MIC data. The identified bacteria derived from the "River: Shinoroshinkawa" site formed a distinct cluster and had higher MIC values than isolates obtained from other locations. Yellow coloring indicates MIC values for isolates derived from "River: Shinoroshinkawa".

## DISCUSSION

Seven distinct locations (Higashitonden, Shinoroshinkawa, Ohno, Shinoro, Tonneusu, Gotenzan, and Yasuharu) around Sapporo City, Japan, were selected as sampling sites. All sampling sites were located within an $\sim$10-km$^2$ area, and demonstrated distinct environmental conditions ranging from natural to urban. Although the sampling sites were geographically close, the prevalence of different damselfly species differed significantly between the sampling sites, with 11 species of damselfly identified across all locations. These differences confirmed the validity of the sampling site selections. In particular, the damselfly population at the "River: Gotenzan" location was unique amongst the sampling sites, with a simple population of *L. sponsa*. This site was located at the highest altitude of all the sampling locations, which may account for the limited population. Thus, our findings agree with previous studies (*Bourret, McPeek & Turgeon, 2012*; *Dolný et al., 2012*; *Dolný, Harabiš & Mižičová, 2014*; *Ball-Damerow, Oboyski & Resh, 2015*) showing that damselflies focally and domestically inhabit a particular location, with minimal migration between sampling sites.

Analyses using SCD-w/o medium revealed a total bacterial load of $\sim$10$^8$ CFU per adult damselfly or larva at all sampling sites, except for "Pond: Ohno". The "Pond: Ohno" site was located upstream of a river and contained little organic matter that would support environmental bacteria, perhaps resulting in a lower bacterial load in insects at this location. Meanwhile, the number of bacteria in larval fecal matter was significantly higher than that in adult damselflies, reflecting the fact that pond and river water contains a much higher level of organic matter that can support fungal prosperity as well as bacterial growth. It is interesting to note that estimation of total bacterial numbers failed in some larval samples with fungal contamination. In addition, because damselflies mature through incomplete metamorphosis (*Abbott & Svensson, 2005*), very few of the gut bacteria present in larvae are transferred to the gut of the adult damselfly. Thus, the fecal bacterial load estimated using SCD-w/o plates can be used as an indicator of damselfly gut bacteria.

We found that changes in environmental temperature were significantly but negatively correlated with total bacterial load on SCD-w/o plates. This may indicate that increases or decreases in temperature causes stress, thereby impairing insect growth (*Leitch & Ceballos, 2008*). This would correlate with the observed decreases in total bacterial numbers in fecal materials with increasing environmental temperature, which was reflected by changes in the gut bacteria. Meanwhile, no significant correlation was found between temperature changes and the antibiotic-resistant bacterial load on SCD-A and MacConkey-A agar plates. It is therefore likely that the prevalence of these bacteria reflects the food intake of the damselflies, which consists of small insects domestically inhabiting each of the sampling locations. In addition, as the cluster analysis based on different indicators (environmental temperature, total bacterial numbers in fecal samples, the prevalence of antibiotic-resistant bacteria) revealed several clades grouped by sampling location, we conclude that these combinations of indicators can be used to characterize traits of each of the sampling locations to determine actual environmental contamination by antibiotic-resistant bacteria.

Thus, damselfly fecal samples can be used to monitor the prevalence of antibiotic-resistant bacteria in local environments.

The predominant bacterial species with antibiotic-resistance isolated from damselfly fecal materials were *Serratia fonticola*, *Serratia liquefaciens*, *Enterobacter aerogenes*, *Enterobacter cloacae*, *Klebsiella oxytoca*, and *Stenotrophomonas maltophilia*, all of which are opportunistic pathogens as causative agents, response for pneumonia, urinary tract infection or septicemia in a compromised host. Although there was no evidence showing any direct contact of these pathogens in damselfly feces to humans on infectious diseases, it is likely that the pathogens have an impact on maintaining human health in urban environments.

Cluster analysis based on MIC data revealed that the identified bacteria derived from the "River: Shinoroshinkawa" site formed a distinct cluster and had higher MIC values than bacteria isolated from other locations. At the present time, the reason for these higher MIC values remains unknown. However, two miniature golf courses are located close to the "River Shinoroshinkawa" sampling site. Historically, large quantities of herbicides and/or pesticides have been used on golf courses to maintain the lawns. It is therefore possible that herbicides or pesticides have induced cross-resistance to antibiotics in environmental bacteria at the "River Shinoroshinkawa" site. This is supported by the fact that acetolactate synthase-inhibiting herbicides and nitrophenolic herbicides can induce cross-resistance to antibiotics in environmental bacterial species with similarities to fungi (*Elanskaya et al., 1998*; *Plaza, Osuna & De Prado, 2003*; *Snelders et al., 2012*). In addition, antibiotic-resistant *Enterobacterales* were only isolated from damselflies, not from larvae. Although the exact reason remains unknown, it is apparent that the flying predation of damselflies is a key factor in the accumulation of antibiotic-resistant bacteria in the gut through the capture of small insects such as flies. Therefore, damselflies are potential vectors in terms of spreading antibiotic-resistant human pathogenic bacteria. Furthermore, animal or birds fed on damselflies would have a physically important role in migrating bacteria inhabited in the gut involving in the spread of *Enterobateriales* seen in damselflies with minimal migration.

## CONCLUSIONS

The present study includes several important findings. First, damselflies focally and domestically inhabit each of the sampling locations, with minimal migration among these areas. Second, the fecal bacterial load, estimated using SCD-w/o medium, can be used as an indicator of damselfly gut bacterial content, thereby reflecting the local ecology at a site. Third, the prevalence of antibiotic-resistant bacteria, estimated using MacConkey-A medium, can be used to monitor antibiotic-resistant bacterial contamination of a location, including *Enterobacterales* strains that are presumably derived from humans. Taken together, we conclude that adult damselfly fecal material can be used as a simple tool to monitor domestic antibiotic-resistant bacterial pollution, particularly those strains associated with human health. However, this study is limited by the small number of environmental samples and the lack of samples from other sources, such as soil. In addition, there is no similar data from other areas that can be used to independently verify our results. Therefore, further studies using additional environmental samples from more diverse locations are needed to confirm our findings.

## ACKNOWLEDGEMENTS

We thank Tamsin Sheen, PhD, from Edanz Group (http://www.edanzediting.com/ac) for editing a draft of this manuscript.

### Funding

This work was supported by grants-in-aid for scientific research from the Japan Society for the Promotion of Science (KAKENHI) [grant numbers 16K15270, 15H05997], and by the Japan Science and Technology Agency. The funders had no role in study design, data collection and analysis, decision to publish, or preparation of the manuscript.

### Grant Disclosures

The following grant information was disclosed by the authors:
Japan Society for the Promotion of Science (KAKENHI): 16K15270, 15H05997.
Japan Science and Technology Agency.

### Competing Interests

The authors declare there are no competing interests.

### Author Contributions

- Yuyu Yamaguchi conceived and designed the experiments, performed the experiments, contributed reagents/materials/analysis tools.
- Torahiko Okubo conceived and designed the experiments, performed the experiments, analyzed the data.
- Mizue Matsushita, Sumio Iwasaki and Kasumi Hayasaka performed the experiments.
- Masashi Wataji conceived and designed the experiments, contributed reagents/materials/analysis tools.
- Kouzi Akizawa and Junji Matsuo contributed reagents/materials/analysis tools, approved the final draft.
- Chikara Shimizu approved the final draft.
- Hiroyuki Yamaguchi conceived and designed the experiments, performed the experiments, analyzed the data, contributed reagents/materials/analysis tools, prepared figures and/or tables, authored or reviewed drafts of the paper, approved the final draft.

### Data Availability

   Raw data are provided in the Supplementary Tables.

### Supplemental Information

Supplemental information for this article can be found online at http://dx.doi.org/10.7717/peerj.5755#supplemental-information.

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
