# Peer review of "Analysis of adult damselfly fecal material aids in the estimation of antibiotic-resistant Enterobacterales contamination of the local environment"

_PeerJ, doi:10.7717/peerj.5755_

## Round 0.1 · original submission · Minor Revisions

Dear Dr. Yamaguchi and colleagues:

Thanks for submitting your manuscript to PeerJ. I have now received two independent reviews of your work, and as you will see, both are very favorable. Well done! Nonetheless, both reviewers raised some relatively minor concerns about the research, and areas where the manuscript can be improved. I agree with the reviewers, and thus feel that their concerns should be adequately addressed before moving forward.

Therefore, I am recommending that you revise your manuscript accordingly, taking into account all of the issues raised by the reviewers. I do believe that your manuscript will be ready for publication once these issues are addressed.

Good luck with your revision,

-joe

Reviewer 1 ·

Basic reporting

The manuscript can be shortened. There are currently unnecessary repetition and redundancies.

Experimental design

Experimental design is basically appropriate.
However, I cannot understand why bacteria grown on SCD agar with an antibiotic were environmental bacteria.

Validity of the findings

This manuscript contains unique and interesting observation concerning antimicrobial resistant bacteria in environments and their monitoring procedure.

Additional comments

Line 38-41: Gut bacteria and environmental bacteria are in reverse order.
Line 250-251: The authors say that "Although there were no obvious differences in the MIC ranges between the sampling sites". However, MIC90 for KAN and CIP of bacteria derived from Shinoroshinkawa is very high. This seems to be obvious.
Figure 5 and Table S1: I cannot understand "SCD-Antibiotics" and "Mc-antibiotics". Do these mean that bacterial growth was observed on agar plate containing either antibiotic? Furthermore, I cannot understand "SCD-MC-all antibiotics".

Reviewer 2 ·

Basic reporting

Need to add more background information especially in introduction section

Experimental design

no comment

Validity of the findings

no comments

Additional comments

Comments:
The paper describes the prevalence of antibiotic-resistant bacteria, including Enterobacterales, in fecal material damselflies (adults and larvae) collected from seven locations around Sapporo City, Japan. This is a well-written manuscript and a thorough and well-designed study. Few typological and grammatical errors are present that need to be corrected. In terms of bacterial identification, a wide range of API identification cut-off value have been considered which usually raise the question of accurate identification of the isolates. It would be therefore highly accepted if it is possible to confirm the bacteria species using 16S rRNA sequencing of the pure isolates.

Introduction
The introduction is comprehensive and well written, provides all the required background.
Line 75-77: “This approach …and urban planners” is not relevant to the context of the study. May delete it.
Line 79-81: Please add more justification why Damselflies could be good indicator of antibiotic resistant bacteria that have potential to infect human and animals.

Materials and methods
Line 95: What was the basis of choosing these antibiotics. Would be great if the can mention it.
Line 99: Delete period (.) after sample collection.
Line 101: what are the distances among all seven locations? I can see it is mentioned in results sections. Please bring it here. Are there any environmental variations among the locations? Please explain it.
Fig 2. “Prevalence of different damselfly species at each sampling site”. Prevalence will not be the right word to use here. What sampling techniques have been used to determine the prevalence? Please explain if do so..
Line 129: Why do incubate at 25o C temperature? Enterobacteriales usually grow at a wide rage of temperature. There might be a possiblily of missing the bacteria that grow above 30o C temperature.
Line 131: Some Enterobacterales are naturally resistant to tetracycline or ampicillin. Therefore, how logical it is to consider these isolates as antibiotic resistant bacteria. Please explain it.
Line 136-137: What cut off value was used to identify the bacterial species using an API 20E kit? Please mention it. As far I can see in table S2, a wide range of cut-off value have been chosen to identify the bacteria which raise the question of accurate identification of the Enterobacteriales species.
Line 149: What alignment technique was used in phylogenetic tree construction? Please mention it.
Line 153-154: “the prevalence of bacteria on SCD-A, the prevalence of bacteria on MacConkey-A”. The authors have not explained how they have determined the prevalence of bacteria on these media plates? Is it prevalence or only presence? Please explain it and correct it throughout the manuscript.

Results:
Line 178-187: I think, this information has already mentioned in materials and methods. Please rewrite the paragraph without repetition of information. Some information might go in materials and methods.
Line 187-193: In my opinion, the author hasn’t determined the prevalence of Damselfly in this study. Therefore, please replace the word “prevalence” with other appropriate words such as “dominant”.

Discussion:
Discussion is clear and well-written. However, the author should focus also on the importance of the Enterobacteriales isolated from the damselfly. The author should add another paragraph describing briefly the importance of these bacteria identified in this study. The author should also describe the potential role of damselfly in spreading human important Enterobateriales as damselflies focally and domestically inhabit areas with minimal migration.

Conclusion:
Conclusion is well-written

Figures and tables:
Table S2: It can be shown in table 2S that many bacterial species have been identified at below 70% API identification cut-off level. I wonder how accurate the identification of these isolates as most of the Enterobacteriales species are closely related. Need more clarification on results.

---

## Round 0.2 · accepted · Accept

Dear Dr. Yamaguchi and colleagues:

Thanks for revising your manuscript based on the minor concerns raised by the reviewers. I now believe that your manuscript is suitable for publication. Congratulations! I look forward to seeing this work in print, and I anticipate it being an important resource for communities studying the impact of the antibiotic-resistant Enterobacterales in local environments, as well researchers on damselflies. Thanks again for choosing PeerJ to publish such important work.

Best,

-joe

#